# Hypophosphatasia: 90 Years from a Canadian Discovery—A Comprehensive Review of the *ALPL* Gene Underlying Rathbun’s Syndrome

**DOI:** 10.3390/genes16121475

**Published:** 2025-12-09

**Authors:** Consolato M. Sergi

**Affiliations:** 1Laboratory Medicine and Pathology, University of Alberta, Edmonton, AB T6G 2R3, Canada; csergi@cheo.on.ca; Tel.: +1-613-737-7600 (ext. 2427); Fax: +1-613-738-4837; 2AP Division, Children’s Hospital of Eastern Ontario, University of Ottawa, Ottawa, ON K1H 8L1, Canada

**Keywords:** bone, hypophosphatasia, soft bones, genetics

## Abstract

Hypophosphatasia (HPP) is an exceptional genetic bone disorder of metabolic character caused by a deficit of the tissue-nonspecific alkaline phosphatase isoenzyme (TNSALP). This protein is encoded by the *ALPL* (alkaline phosphatase liver/bone/kidney) gene. In the medical literature, HPP is also known as Rathbun’s syndrome, named after the Canadian physician who first identified this disorder. Patients exhibit persistently low serum alkaline phosphatase (ALP) levels. In fact, ALP renders this measure a reliable indicator of the condition. Adult HPP is varied, with some patients exhibiting only moderate, non-pathognomonic symptoms. They include arthropathy, arthrodynia, chondrocalcinosis, osteopenia, osteomalacia, and generic musculoskeletal discomfort. Healthcare may require coordinating several services to manage a patient with HPP. This comprehensive review will highlight the genetic knowledge, pathology data, and patient management approaches, including Medicare’s coverage. In addition, this paper aims to address specific themes related to HPP, including its significance, current challenges, and controversies.

## 1. Introduction

Hypophosphatasia (HPP) is a condition marked by decreased blood and bone alkaline phosphatase (ALP) activity, inadequate mineralization of growing or remodeling bone, and tooth loss despite intact roots. In severe cases, this illness manifests as stillbirth with no mineralized bone; in less severe cases, it manifests as the development of pathological fractures in the lower extremities during maturity. This paper aims to address specific themes related to HPP. What is the significance, current challenges, and controversies of hypophosphatasia? First, HPP’s primary importance lies in its impact on bone and teeth mineralization due to a deficiency in the tissue-nonspecific alkaline phosphatase (TNSALP) enzyme, which, if untreated in severe cases, can be fatal. HPP is significant because it affects multiple systems, leading to severe outcomes, particularly in infants. Perinatal and infantile forms can cause profound skeletal demineralization, pulmonary hypoplasia, and vitamin B6-dependent seizures, resulting in high mortality without treatment. There is a significant morbidity across ages, with even milder forms in children and adults causing significant chronic pain, recurrent fractures, osteomalacia (bone softening), muscle weakness, and premature tooth loss, significantly impacting quality of life and potentially leading to severe disability. TNSALP’s crucial role in hydrolyzing inorganic pyrophosphate (PPi), an inhibitor of hydroxyapatite formation, highlights the importance of this pathway in normal bone development. Its deficiency disrupts the entire mineralization process. The development and approval of asfotase alfa, an enzyme replacement therapy (ERT), represents a significant advance, transforming the prognosis for severe cases and offering substantial improvements in bone health, physical function, and survival. This paper provides an update on this disease, also considering its significance, current challenges, and controversies.

## 2. HPP Classification

Based on the gravity of its symptoms and the primary age at diagnosis, HPP is frequently divided into seven clinical groups and one anomalous cluster [1,2,3,4,5,6,7,8,9].
**Perinatal (“*severe*”) HPP** refers to a situation where the body has high calcium levels and inadequate lung function. The most severe type, known as perinatal severe (lethal) hypophosphatasia, is characterized by the development of severe symptoms that are frequently identified by ultrasonography before birth. The onset occurs in utero or at birth. Significant bone mineralization is a primary characteristic, resulting in short, bent, and underdeveloped limbs. Other factors include a flail chest, which is an irregularly formed chest, and severely underdeveloped ribs, which can cause life-threatening respiratory failure. In utero or in the days or weeks following delivery, the death rate is significant [1,4,5,9,10,11,12,13,14].**Perinatal (“*mild*”) HPP** includes a disease exhibiting skeletal symptoms that appear during pregnancy and gradually improve, leading to one of the less severe forms. Although less severe, this type is also seen during pregnancy. Later in pregnancy or after delivery, the condition typically improves on its own. It starts in utero. Skeletal anomalies, such as the bowing of long bones, are key characteristics that eventually resolve on their own. Later, the patient develops a milder form of the illness that resembles HPP in children or adults [1,9,15,16,17,18].**Infantile HPP** is characterized by an absence of a rise in serum ALP activity, and clinical signs of rickets begin during the first six months after birth. This type of symptom usually manifests during the first half year of a baby’s life. Failure to flourish, poor eating, muscular hypotonia, cranial synostosis with resulting elevated intracranial pressure, rickets-like skeletal deformities, early loss of primary (baby) teeth, and ultimately seizures responsive to vitamin B6 are some of the key characteristics [1,8,9,12,18,19,20,21,22].**Childhood/Juvenile HPP (mild–severe)** is characterized by clinical manifestations occurring in the 1st and 2nd decades after infancy. The severity of childhood hypophosphatasia varies greatly, ranging from moderate to severe, and it manifests after six months of age. One of the main characteristics is (1) the early loss of primary teeth, which often occurs before age five; (2) rickets-related skeletal abnormalities, such as small stature and bent legs; (3) muscle weakness and bone and joint pain; and (4) a waddling gait and delayed motor skills [4,12,15,22,23,24,25,26,27,28,29,30,31,32].**Adult HPP** affects patients who may have experienced milder symptoms as children, but this variety usually manifests in middle life. The following are essential characteristics: (1) Osteomalacia (“softening of the bones”), which causes frequent stress fractures, particularly in the thighs and feet; (2) early adult tooth loss; and (3) joint inflammation and discomfort, as well as persistent musculoskeletal pain [4,10,20,33,34,35,36,37,38,39,40,41,42,43,44,45,46].**Odontohypophosphatasia** is categorized by a lack of any skeletal symptoms. This variant is distinguished by the early loss of primary teeth and/or severe dental caries. This is the mildest type of HPP, with no skeletal problems and only tooth damage. At any age, it can happen. Abnormal tooth growth and early loss of primary or adult teeth are essential characteristics. Dental caries may be present, and the roots of the missing primary teeth are frequently still intact [7,8,10,12,47,48].

It is necessary to discuss another variant, “Pseudohypophosphatasia”, which is infrequent. Although the patients’ serum ALP levels are normal or even elevated, they display the clinical and radiographic signs of infantile HPP. It is similar to HPP in infants. However, unlike all other types of HPP, this disorder is distinguished by a normal or elevated serum ALP amount [10,49,50,51,52,53].

## 3. Rathbun and Whyte

Before disclosing the radiologic and pathological features of this syndrome, it is opportune to delve into history. Initially described in 1936, it was comprehensively named and subsequently documented by Canadian doctor John Campbell Rathbun (1915–1972) during the examination and treatment of a male infant with very low ALP levels in 1948 (Figure 1).

The genetic foundation of the disease was delineated approximately 40 years later. HPP is occasionally referred to as Rathbun’s syndrome, named after its primary documenter. Briefly, John Campbell Rathbun was born on 8 May 1915 in Toronto, ON, Canada. His education began at Upper Canada College in 1928 and continued until 1933. He received the title of Doctor of Medicine at the University of Toronto in 1939. He practiced medicine and specialized in pediatrics in London, ON, Canada, from 1948 to 1972. He was a professor and head of the department of pediatrics for almost two decades. He served as a lieutenant commander in the English Royal Navy and the Royal Canadian Navy. His legacy remains tangible in Ontario and unscathed by the two world wars and Canadian cancel culture (John Campbell Rathbun (8 March 1915–31 October 1972), Canadian educator, physician | World Biographical Encyclopedia). Rathbun’s studies on HPP are also mirrored in the splendid works of Étienne Mornet and Michael Whyte. Dr. Whyte’s legacy is probably unparalleled in the field of medicine. Michael Whyte, MD, from Washington University in St. Louis, has conducted research on HPP, focusing on its clinical aspects, including the natural history and treatment responses. Dr. Michael Whyte received his MD from Downstate College of Medicine, the State University of New York, in Brooklyn, NY, and completed an internal medicine residency at Bellevue Hospital in New York City. In the late 1970s, he joined the faculty after his endocrinology fellowship. He directed the Center for Metabolic Bone Disease & Molecular Research located at Shriners Hospitals for Children in St. Louis for nearly 40 years. His studies examined the relationship between HPP and high levels of pyridoxal 5′-phosphate (PLP), the primary (circulating) form of vitamin B6.

Dr. Whyte’s work also suggested that HPP can indeed involve elevated serum vitamin B6 levels, which may contribute to neuropathic pain in some affected individuals.

## 4. Epidemiology

According to some statistics from Ontario, Canada, the birth prevalence of perinatal/infantile HPP is estimated to be 1 in 100,000 births [56]. The Hardy–Weinberg equation (HWE) suggests that the prevalence of individuals carrying both copies of the pathogenic mutations is approximately 1 in 150. The perinatal (severe) variant of the p.Gly334Asp mutation occurs in approximately 1 in 2500 individuals among the Canadian Mennonite population, exhibiting a carrier frequency of approximately 1 in 25. Molecular diagnostic testing in France and other European nations estimates the prevalence of severe variants at 1 in 300,000. The prevalence of mild forms of HPP (perinatal benign, mild childhood, adult, and odonto-HPP/odontohypophosphatasia) is estimated at 1 in 6300 individuals [57,58]. This occurs because individuals who possess a single copy of the defective gene (heterozygotes) may still exhibit the disease, albeit with reduced severity. The application of the HWE to this estimation for severe forms suggests that the likelihood of an individual carrying one copy of the ALPL pathogenic variant in France is approximately 1 in 275. The incidence of severe HPP at birth in Japan is believed to be 1 in 150,000. This estimate relies on the prevalence of individuals possessing two copies of the pathogenic variant c.1559delT, quantified at 1 in 900,000 [39], and the percentage of affected persons of Japanese heritage who exhibit this variant, recorded at 45.4%. In China, specific deleterious mutations have been recorded [59,60,61]; however, the precise incidence at birth remains unspecified. No instances of HPP have been reported in the medical literature of Africa, at least according to our records, except in some regions in North Africa (French-speaking) and South Africa (English-speaking). It is crucial to acknowledge the significant clinical ascertainment bias present. HPP is also infrequently observed in African American individuals; it is believed that the presence of pathogenic genetic variants in these demographics results from admixture with European ancestry. Selected epidemiological data on severe HPP are presented in Table 1.

Milder forms of HPP, which typically appear in childhood or adulthood, are more frequent than the severe forms and are often underdiagnosed due to non-specific symptoms such as chronic pain and recurrent fractures. There is a significant diagnostic delay, particularly in adults, reflecting limited awareness among healthcare professionals. This suggests that the actual prevalence of the milder forms may be higher than current estimates.

## 5. Laboratory Diagnosis

HPP can be diagnosed in an individual who exhibits symptoms, laboratory results, and radiographic features consistent with the condition. Diagnosis is confirmed by measuring low blood ALP activity and/or identifying specific *ALPL* gene mutations. These mutations can produce a loss of function in both copies of the gene or in a single mutation with a dominant-negative effect.

Laboratory characteristics include hypercalciuria, especially in early infancy, with or without hypercalcemia, and normal serum and ionized calcium levels. It may be high, especially in the first year. Usually, normal serum and urine inorganic phosphate levels are observed. Serum 25-hydroxy and 1,25-dihydroxyvitamin D as well as the level of parathyroid hormone are normal, but PLP, a vitamin B6 metabolite, is elevated.

Vitamin B6-containing multivitamins or calcium supplements taken one week before serum PLP testing may cause false-positive results. In addition, the urine amino acid chromatogram shows elevated levels of phosphoethanolamine (PEA) and proline. Other metabolic bone diseases may promote urinary PEA. Patients may have normal urine PEA levels, while asymptomatic heterozygotes may have elevated amounts. The urine also has more PPi. Asymptomatic patients with heterozygote status may harbor high urine PPi, and, finally, decreased serum unfractionated ALP. Serum enzymatic activity may rise after pregnancy, liver disease, acute fractures, and surgery. Thus, idiopathic fractures in children may require repeated evaluations. Serum bone isoform of ALP activity may be needed in case of liver illness. The liver isoenzyme is thermally stable, although the bone isoenzyme is thermolabile. Asymptomatic heterozygotes may have low serum ALP.

## 6. Genetics

The *ALPL* gene encodes tissue-nonspecific alkaline phosphatase (TNSALP). As seen above, this isozyme is found in the liver, kidneys, and bones. It functions as a homodimer. The enzyme functions as a membrane-bound phosphatase. The natural substrates include PPi, PLP, and PEA. Pathogenic mutations in *ALPL* are distributed evenly across the 12 exons. Pathogenic missense mutations account for 74.6% of all variations. The remaining variants include microdeletions and insertions (13.3%), rare pathogenic splice-site variants (6.0%), ultrarare pathogenic nonsense variants (3.7%), significant deletions (1.3%), and nucleotide alterations impacting the major transcription initiation site. The correlation between genotype and phenotype has been examined using site-directed mutagenesis and three-dimensional modeling of the enzyme. The previously described investigations have permitted the identification and categorization of alleles into two classifications: severe and mild or moderate. Severe alleles denote those that cause a substantial decrease in enzyme activity, whereas moderate alleles maintain a degree of residual enzymatic function. Moreover, specific alleles exert a dominant-negative effect, leading to dominant inheritance. The studies encompass the research conducted by numerous authors [62,63,64,65,66,67,68,69,70]. Nonetheless, these techniques do not consistently foretell the severity of HPP pathogenic variants with high precision. Pathogenic mutations may result in many outcomes that might occasionally aggregate. The repercussions include a reduction or complete loss of catalytic enzymatic activity, an intrinsic inability to create homodimers, and the entrapment of mutant proteins within cellular compartments, hindering their transit to the cell membrane [62,63,66,68,71,72,73].

Genetic counseling provides insights into the characteristics, inheritance patterns, and pertinent consequences of genetic abnormalities, enabling people and families to make informed choices regarding their health and personal matters. Autosomal dominant inheritance is linked to dominant-negative *ALPL* gene mutations. Family clinical variability is common, especially when some family members have heterozygous pathogenic *ALPL* mutations and others have biallelic variants. Families with two *ALPL* pathogenic mutations may have severe perinatal, childhood, and adult HPP. The proband’s genetic mutation(s) must be identified to estimate the likelihood of disease recurrence. Molecular testing on the proband’s parents is also necessary to establish their proper status. If a pathogenic genetic mutation is identified in only one parent, and biological parentage has been validated through parental identity testing, it is conceivable that one of the disease-causing mutations present in the affected individual arose spontaneously in that individual (termed a de novo event) or in one of the parents after fertilization (referred to as a postzygotic event [de novo] in a mosaic parent). This hypothesis is corroborated by research undertaken by Taillandier et al. in 2005 and Zhang et al. in 2012 [60,74]. If the subject exhibits two identical deleterious genetic variants, alternative options to consider include: (1) No loss in the proband, whether single-exon or multi-exon, was detected using sequence analysis. This deletion created a misleading impression of homozygosity. (2) The proband displayed homozygosity for the fatal mutation resulting from uniparental isodisomy of the paternal chromosome. In the Canadian Mennonite population, which resides mostly in central Canada, the incidence of the severe perinatal form is 1 in 2500, while the carrier frequency is 1 in 25. This is attributed to a specific genetic variation that originated from a common ancestor. Each sibling of the proband’s parents has a 50% probability of inheriting a heterozygous pathogenic mutation in the *ALPL* gene. Prior identification of the *ALPL* pathogenic variants within the family is essential for heterozygote testing of at-risk relatives. Individuals diagnosed with HPP due to a heterozygous ALPL variant exhibiting a dominant-negative effect have inherited the pathogenic *ALPL* mutation from a parent. The parents may present clinical indications of HPP or may not. Evaluating the parents of a proband necessitates a review of their medical history and the performance of laboratory testing to detect any signs of HPP. If an individual’s physical traits resemble those of many skeletal dysplasias, it is prudent to pursue complete genomic testing. This type of testing does not necessitate the clinician to pinpoint the individual gene that may be involved and is regarded as the optimal option. The predominant technique utilized is exome sequencing, although genome sequencing is also viable. Next-generation sequencing (NGS) is a valuable tool in the laboratory diagnosis of HPP, primarily by improving efficiency, enabling comprehensive differential diagnosis, and identifying novel or rare pathogenic variants. The diagnosis of HPP relies on clinical features, biochemical tests (low ALP), and DNA analysis of the *ALPL* gene. NGS enables simultaneous sequencing of the *ALPL* gene and other genes involved in conditions that present with similar symptoms (e.g., osteogenesis imperfecta or other forms of rickets). This multi-gene panel approach facilitates efficient differential diagnosis, which is crucial due to the significant phenotypic overlap between HPP and other rare genetic diseases [29,75,76,77,78,79,80,81,82,83,84,85,86,87]. While traditional Sanger sequencing can detect variants in approximately 95% of HPP cases, some variants in deep intronic or regulatory sequences, or large deletions/duplications, might be missed. NGS, especially with supplemental analyses, helps identify less common or novel variants, thereby increasing overall diagnostic yield and reducing the number of unclassified cases. Traditional gene-by-gene analysis using Sanger sequencing is expensive and time-consuming. NGS can sequence multiple genes in parallel, significantly reducing turnaround time for a conclusive diagnosis. This enables timely and appropriate patient management, which is critical for preventing misdiagnosis and potentially harmful treatments (e.g., bisphosphonates in HPP patients). Identifying specific *ALPL* gene variants through NGS helps establish genotype-phenotype correlations, which can assist clinicians in predicting disease severity, understanding inheritance patterns (e.g., dominant-negative effects), and providing more accurate prenatal and postnatal counseling. Moreover, NGS can help resolve complex cases in which initial biochemical results are inconclusive or clinical presentations are atypical (e.g., in adults with mild, nonspecific symptoms). In summary, NGS offers a more efficient, comprehensive, and accurate approach to genetic analysis of HPP than traditional methods, thereby improving patient care and outcomes.

## 7. Imaging

Prenatal ultrasonography is frequently employed to detect severe perinatal HPP (Figure 2). Certain pregnancies may culminate in stillbirth. Both stillborn and live-born neonates display a reduced thoracic cavity and limbs that are short and curled.

There can be a fluttering chest. Babies born with perinatal HPP may have pulmonary insufficiency, the most common cause of death in this condition being restrictive lung disease. Apnea and convulsions sometimes go hand in hand with hypercalcemia. A unique phenotype, termed “treated perinatal and infantile HPP,” has been identified in people receiving asfotase alfa ERT. Nonetheless, even with timely diagnosis, there remains a possibility of encountering adverse outcomes with ERT [88]. Infants diagnosed with perinatal (severe) HPP, who received ERT during the first day to 78 months of age, showed improvements in pulmonary function and elevated survival rates. The effect of ERT on fractures is still ambiguous, as noted by Whyte et al. in 2019 [13,89]. Historically, individuals with severe physical traits died before their teeth emerged; recent research suggests that neonates undergoing ERT may exhibit dental characteristics.

Perinatal (benign) HPP is classically identified through prenatal ultrasound, which demonstrates shorter and bent long bones, although mineralization is observed to be normal or slightly diminished. Postnatally, the skeletal manifestations progressively improve, resulting in a milder variant of HPP [90]. Conducting an ultrasound examination in the early stages of pregnancy might facilitate the assessment of diseases such as “osteogenesis imperfecta” (OI) type II, skeletal chondrodysplasias with variable bony mineralization abnormalities, and HPP, in addition to campomelic dysplasia (“bent limb”), a severe, 17q24 chromosomally located, *SOX9*-gene associated, life-threatening genetic disorder characterized by evident bowing of the long bones, a small and narrow chest, and various other skeletal and extraskeletal abnormalities. Proficient sonographers generally encounter few difficulties in distinguishing between these conditions. Fetal radiographs help detect diminished bone mineralization, a hallmark of perinatal HPP, which distinguishes it from other disorders within the differential diagnosis.

At birth, differentiation among OI type II, thanatophoric osteodysplasia, campomelic skeletal dysplasia, and chondrodysplasias exhibiting bony mineralization defects, as well as HPP, is readily achievable through radiographic evaluation (see below for details). For patients with an unclear diagnosis, assessing serum ALP, the level of PLP, or, specifically, vitamin B6, as well as urine PEA, can provide diagnostic insights until validation is reached via molecular biology (genetic) testing.

In infancy, childhood, and youth, the infantile variant of this illness manifests with symptoms including irritability, inadequate eating, failure to thrive, hypotonia, and seizures. These symptoms may suggest alternative diseases, such as congenital energy metabolism disorders, organic acidemia, secondary and primary rickets, child neglect, and non-accidental bony injuries. Consequently, a thorough assessment is essential to ascertain the root cause.

## 8. Critical Differential Diagnosis

The cruciality of the differential diagnosis of HPP cannot be understated. It includes conditions that cause skeletal deformities and mineralization defects, such as OI, campomelic dysplasia, achondrogenesis, and various forms of rickets/osteomalacia resulting from vitamin D deficiency or hereditary phosphate-wasting conditions. Other considerations are renal osteodystrophy, fibrous dysplasia, and tumor-induced osteomalacia, with the specific differential diagnosis depending on the patient’s age and clinical presentation.

Conditions presenting with HPP-similar skeletal defects can be further characterized here:(1)OI: A group of genetic disorders characterized by brittle bones, which are a common differential diagnosis for severe forms of HPP.(2)Campomelic Dysplasia: Another skeletal dysplasia that can present similarly to severe HPP, requiring genetic testing for diagnosis.(3)Achondrogenesis: A severe form of chondrodysplasia that can be difficult to distinguish from severe perinatal HPP, particularly prenatally.(4)Chondrodysplasias to be further specified: It is a broad category of disorders affecting cartilage and bone development, which can have overlapping features with HPP. Other bone disorders include conditions like cleidocranial dysostosis and Cole-Carpenter syndrome, which can also be considered, especially in childhood. Metabolic conditions with bone involvement include rickets and osteomalacia. This encompasses cases of rickets due to vitamin D deficiency, hereditary hypophosphatemic rickets, and tumor-induced osteomalacia, all of which present with impaired bone mineralization.(5)Renal osteodystrophy: Bone disease that occurs in people with chronic kidney disease, which can manifest with skeletal issues similar to HPP.(6)Other potential diagnoses include fibrous dysplasia of bones, a condition where normal bone is replaced by fibrous tissue, leading to bone deformities, and Renal Fanconi Syndrome, a disorder of the kidney tubules that can also lead to bone disease [91].

## 9. Therapeutic Strategies

Research has shown that asfotase alfa (Strensiq^®^), an ERT, improves respiratory function, calcium ion homeostasis, skeletal comprehensive health, and overall survival rates in patients with infantile and early childhood (juvenile) HPP [16,29,41,79,83,92,93,94,95,96,97,98,99,100,101]. ERT has seen a rising use in individuals with the severe perinatal form, alongside a growing utilization of ERT for the treatment of osteomalacia in adults. The management of severe perinatal cases requires supportive care. This involves overseeing the patient’s status through expectant management and aiding the family. Respiratory assistance is administered, while an endocrinologist and orthopedist must supervise the patient’s calcium balance and bone health. Pain treatment is employed, and craniosynostosis is addressed with neurosurgical intervention. A nephrologist oversees kidney disease, and dental treatment is administered.

The recommended treatments for infants and young children include respiratory support, calcium and bone health regulation under the supervision of an endocrinologist and an orthopedist, pain management, vitamin B6 administration for seizure treatment, surgical intervention for craniosynostosis, kidney disease management by a nephrologist, and dental care. Dental care for any other conditions should commence at age 1. Nonsteroidal anti-inflammatory drugs are indicated for osteoarthritis, bone discomfort, and osteomalacia. Internal fixation is advised for skeletal pseudofractures and stress fractures. “Teriparatide”, a potent anabolic (bone-building) medication primarily used to treat severe osteoporosis in patients who are at a high risk of bone fractures, has shown limited application in the management of osteomalacia for people harboring HPP [17,102].

In HPP monitoring, particular emphasis should be placed on evaluating and analyzing calcium homeostasis and skeletal health under the guidance of endocrinologists, nephrologists, and orthopedists. It is essential to perform assessments by specialists in physical medicine and rehabilitation, physical therapists, and occupational therapists as required. It is also vital to schedule nephrology assessments as needed for renal disease and neurology assessments as required for seizures. It is also critical to arrange dental appointments biannually, commencing at the age of one. It is worth considering that teriparatide is contraindicated for children due to the presence of bisphosphonates and elevated vitamin D levels. Moreover, the implications of providing asfotase alfa (Strensiq^®^, Alexion Pharmaceuticals, Inc., Boston, MA, USA) ERT to pregnant women have not been comprehensively studied, leaving the potential risks to the growing fetus ambiguous. This may be addressed in future prospective experimental animal and clinical studies [103].

As indicated above, HPP constitutes a relative contraindication for bisphosphonate administration. Broadly, no or only mild adverse effects have been reported in children with the infantile variant [104]. Still, there remains a persistent theoretical concern about the composition of bisphosphonates. The phosphate configurations observed in bisphosphonates closely mirror those in PPi, the natural substrate for TNSALP. Consequently, it is posited that bisphosphonate treatment can be compared to “adding fuel to the fire.” Lateral subtrochanteric femoral pseudofractures have been documented in individuals with HPP and osteomalacia following bisphosphonate treatment [105,106]. The prevalence of adult HPP is uncertain, and many untreated individuals are presumably receiving bisphosphonates, rendering the incidence of this rare outcome similarly unknown. Excessive consumption of vitamin D can aggravate hypercalcemia/hypercalciuria in infants with infantile HPP who already exhibit hypercalcemia. Elevated doses of the drug “teriparatide”, a recombinant fragment of human parathyroid hormone including amino acids 1–34, have been associated with the induction of osteosarcoma in rats [107,108,109]. Moreover, there exists a potential heightened risk of radiation-induced osteosarcoma, a tumor that impacts the growth plate in children, among individuals. Consequently, its usage in children with HPP is not advised. The impact of asfotase alfa (Strensiq^®^) ERT on human gestation has not been thoroughly examined. Therefore, the potential risk to the developing fetus during this treatment for a pregnant woman remains ambiguous. It has been suggested that families consult the website “MotherToBaby,” or their local obstetricians/midwives or general practitioners, for more detailed information on the use of medication during pregnancy.

Augmentation of osteoblasts with the use of antibodies directed against sclerostin. Teriparatide enhances osteoblast activity by increasing TNSALP production, whereas sclerostin inhibits osteoblast growth. Metabolic bone disorders can now be addressed with anti-sclerostin therapies. A Phase II open-label clinical trial was conducted in eight individuals with HPP, with an average age of 47.8 years. The trial employed anti-sclerostin monoclonal antibodies (BPS804 [110]). The findings indicated that seven participants who completed the 16-week study period demonstrated early improvements in bone density and markers of bone turnover. A 2003 case report by Whyte et al. illustrated the practical application of hematopoietic cell transplantation, also known as bone marrow transplantation, for treating severe HPP in an eight-month-old female patient [111]. The surgery yielded substantial and enduring clinical and radiologic enhancement. Seven years after her transplant, she was still strong [112]. Nonetheless, a neonate can develop leukemia because of treatment [113]. The administration of ex vivo-expanded mesenchymal stem cells has been shown to improve bone mineralization, muscle mass, respiratory function, cognitive development, and survival in individuals who had previously undergone bone marrow transplantation [114].

## 10. Critical Current Analysis and Conclusive Remarks

Overall, HPP is a diagnosis that cannot be delayed or missed. As seen, current challenges lie ahead. They include delayed diagnosis and misdiagnosis. In fact, the diagnosis is often significantly delayed, particularly in adults, with a median delay of 5.7 years from symptom onset reported in one registry. Symptoms like chronic pain and fractures are often misattributed to more common conditions like osteoporosis or fibromyalgia, leading to inappropriate treatments that can worsen HPP. The wide variability in how the disease presents (phenotypic heterogeneity) makes it difficult to establish clear genotype-phenotype correlations and challenging for clinicians to recognize, as symptoms can overlap across different age-of-onset classifications. Thus, updated continuing reviews of HPP are critical for optimal health care and physician education. Moreover, while ERT improves skeletal and respiratory function, management of other multi-system manifestations, such as chronic pain, neuropsychiatric symptoms (e.g., fatigue, depression, anxiety), and dental issues, requires a complex, multidisciplinary approach. Access to ERT can be limited by high cost and strict approval criteria, which may only cover the most severe pediatric-onset cases in some regions, leaving many symptomatic adults without a licensed treatment option. Medicare does not automatically cover asfotase (Strensiq^®^), as it is a specialty drug with high costs, and Medicare’s coverage of outpatient medications depends on having a Part D plan or a “Medicare Advantage plan” with drug coverage. Coverage decisions are made on a case-by-case basis. They are typically dependent on factors like the specific diagnosis (e.g., pediatric-onset HPP), the patient’s insurance plan, and the product’s approval status. Access to treatment almost always relies on reimbursement programs, and patients should check their specific Medicare plan to see if it covers treatment. Clear, evidence-based guidelines for initiating ERT in adults with HPP are still emerging, creating a “gray zone” for treatment decisions in patients with intermediate severity. A major controversy is whether all symptomatic adults, or only those with milder forms (such as odonto-HPP), should receive ERT. While some studies show benefit, the cost-effectiveness and long-term risk-benefit ratio, especially regarding potential ectopic calcifications, remain debated. Limited evidence suggests that abruptly stopping ERT can lead to rapid clinical deterioration, underscoring the importance of continuous, potentially lifelong, treatment once initiated. This raises ethical and practical questions about long-term management and the need for robust data on discontinuation effects in adult patients. In conclusion, there is probably still a lot of intense work ahead, and screening programs in particular regions of the world (such as Manitoba, Canada) need to be intensified. The broader adoption of NGS will be critical for the future, and, finally, the role of school inspectors and periodic medical visits may be emphasized for some geographical areas.

## Figures and Tables

**Figure 1 genes-16-01475-f001:**
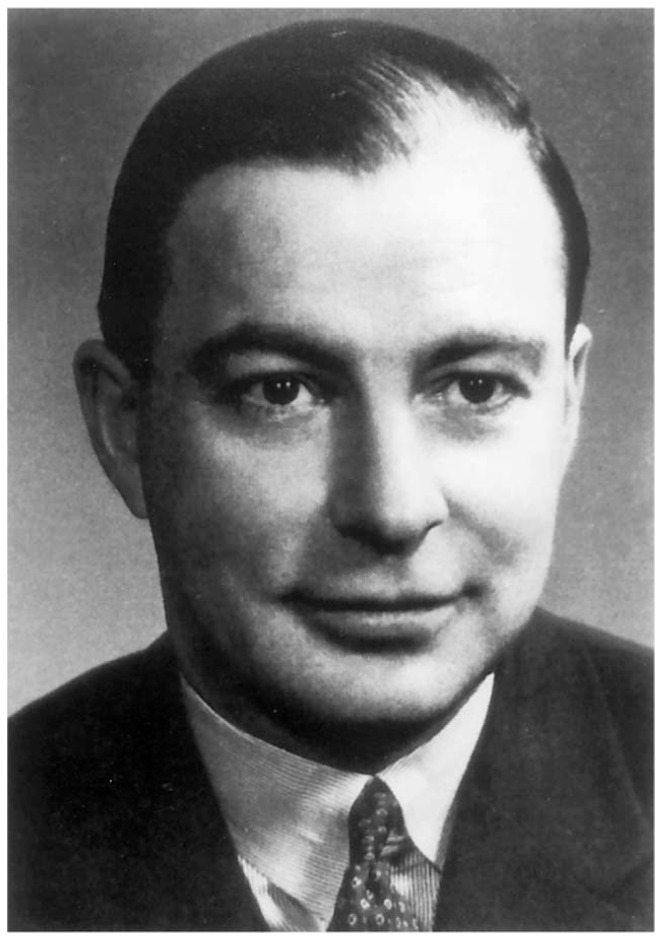
Canadian physician John Campbell Rathbun (1915–1972), photograph. This photograph originates from a freely accessible scientific article by Mumm et al., J Bone Miner Res. 2001 Sep;16(9):1724–7 [54]. In this open-access article, the photograph of Dr. John Campbell Rathbun (1915–1972) is shown. His breakthrough of hypophosphatasia occurred in 1948 (the original picture was reproduced in an open-access journal with historic permission from MP Whyte 1994 [55]. Endocr Rev 15:439–461).

**Figure 2 genes-16-01475-f002:**
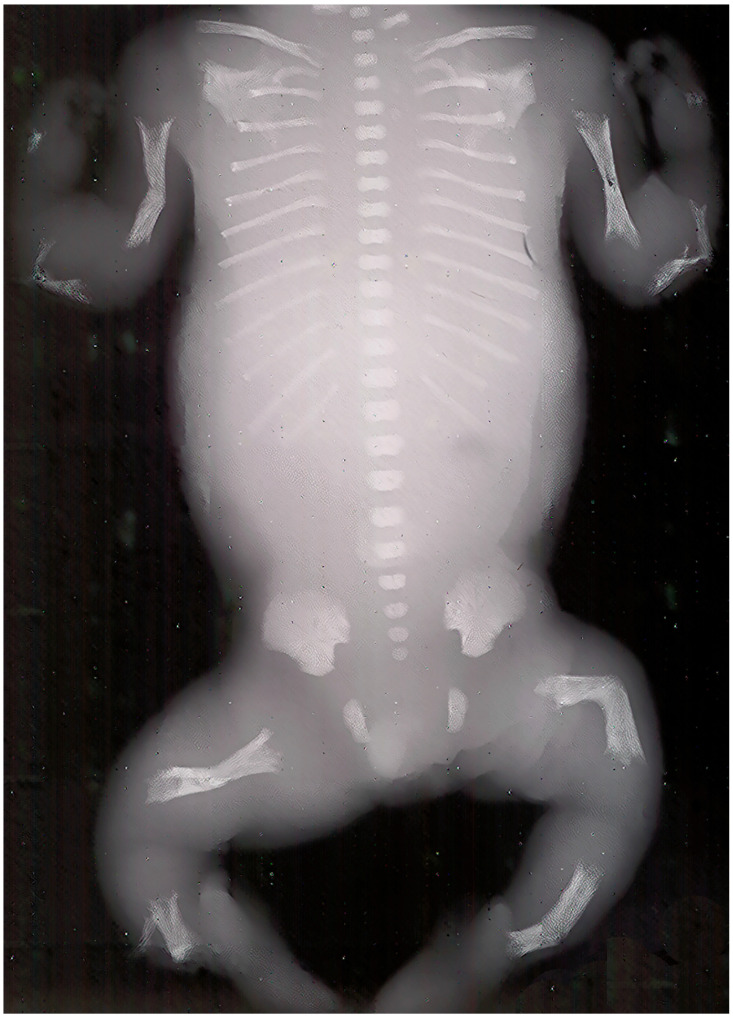
Perinatal X-ray showing the characteristic radiological features of hypophosphatasia. This photograph originates from the original X-ray of one of the author’s patients (Personal archive)—no copyright issues.

**Table 1 genes-16-01475-t001:** Severe HPP Selected Epidemiology.

Population/Region	Estimated Prevalence (Live Births)
Mennonite Canada (MB, Canada)	1 in 2500
Canada (general population)	1 in 100,000
Europe	1 in 300,000
Japan	1 in 150,000 to 1 in 500,000
China	Unknown *

Note. In Europe, the estimated prevalence of mild forms ranges from 1 in 2430 to 1 in 6370 people. * In China, it is unknown what the estimated prevalence is, but cases have been reported.

## Data Availability

No new data were created or analyzed in this study. Data sharing does not apply to this article.

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
