# Peer review of "Hypophosphatasia: 90 Years from a Canadian Discovery—A Comprehensive Review of the APLP Gene Underlying Rathbun’s Syndrome"

_genes, 2025, doi:10.3390/genes16121475_

Round 1

Reviewer 1 Report

Comments and Suggestions for Authors

1. Perhaps it would be worth adding statistics for various populations, based on various open genomic projects.
2. Line 223 - c.1559delT – for all mutations, the rsID should also be added for more precise labeling.
3. Incorrect reference to Figure 2A (should be a reference to Figure 3) – line 437.
4. There is no description of the second figure in the text (with the protein structure, and it is also unclear what the semantic meaning of this figure is).
5. The description of diagnostic methods lacks a modern diagnostic algorithm (for example, there are methods for prescribing NGS after laboratory diagnostics). Having such a recommended approach would significantly increase the value of the article.

Author Response

  1. Perhaps it would be worth adding statistics for various
    populations, based on various open genomic projects.
    2. Line 223 - c.1559delT – for all mutations, the rsID should also be
    added for more precise labeling.
    3. Incorrect reference to Figure 2A (should be a reference to Figure
    3) – line 437.
    4. There is no description of the second figure in the text (with the
    *
    Comments 1: [Paste the full reviewer comment here.]
    Response 1: [Type your response here and mark your revisions in
    red] Thank you for pointing this out. I/We agree with this
    comment. Therefore, I/we have….[Explain what change you have
    made. Mention exactly where in the revised manuscript this change
    can be found page number paragraph and line ]“[updated text
    protein structure, and it is also unclear what the semantic meaning
    of this figure is).
    5. The description of diagnostic methods lacks a modern diagnostic
    algorithm (for example, there are methods for prescribing NGS
    after laboratory diagnostics). Having such a recommended
    approach would significantly increase the value of the article.

Thank you for the opportunity to revise our manuscript and thank you for your suggestions and comments.

  1. Perhaps it would be worth adding statistics for various
    populations, based on various open genomic projects.
  2. Yes, we added a table with some statistics.

  3. 2. Line 223 - c.1559delT – for all mutations, the rsID should also be
    added for more precise labeling.
  4. Thank you, Yes, we changed it.
    3. Incorrect reference to Figure 2A (should be a reference to Figure
    3) – line 437.
  5. Figures 2 and 3 have been deleted, because the knowledge is publicly available as requested by another reviewer.
    4. There is no description of the second figure in the text (with the
    *
    5. The description of diagnostic methods lacks a modern diagnostic
    algorithm (for example, there are methods for prescribing NGS
    after laboratory diagnostics). Having such a recommended
    approach would significantly increase the value of the article.

Yes, thank you. A specific paragraph on NGS is provided.

Thank you again for your comments.

Reviewer 2 Report

Comments and Suggestions for Authors

Hypophosphatasia: 90 Years from a Canadian Discovery. A  Comprehensive Analysis of the APLP Gene Underlying Rathbun's Syndrome.

 An interesting review manuscript summarizing what is relevant for hypophosphatasia, important from both clinical practice as well as research point of view.  

Introduction- is comprehensive, with a detailed presentation of the main symptoms of each type of hypophosphatasia.

Diagnosis- described extremely accurate the main methods as well as the diagnosis symptom based.

Epidemiology- described the various levels of prevalence in both North American continent and Europe.

Genetics accurately described diagnosis-based enzymes.

Imagining- displayed a suggesting image of the bone problems.

Differential diagnosis- appropriate

Therapeutic strategies-described the main relevant therapeutical directions.

Reference is relevant

Author Response

Thank you for your comments and suggestions. Please revise the manuscript accordingly, including those of the other reviewers.

Thank you again for your time in peer-reviewing this manuscript.

Reviewer 3 Report

Comments and Suggestions for Authors

The manuscript presents a comprehensive overview of hypophosphatasia and the ALPL gene, including inheritance patterns, genetic counseling, diagnostic approaches, imaging, and therapeutic strategies. While the paper demonstrates significant effort and includes extensive background information, it does not meet the structural, analytical, and critical standards expected of a publication in a peer-reviewed scientific journal.

Introduction - it does not identify the specific problem or gap that the paper seeks to address. Instead, it gives information about the clinical presentation. The introduction should clearly outline the significance of the topic, current challenges or controversies, and the purpose and scope of the paper.

Rathbun and Whyte section - I appreciate the author’s effort to acknowledge the seminal contributions of Dr. John Rathbun and Dr. Michael Whyte to the understanding of hypophosphatasia. However, this section is overly detailed and reads more like a biographical narrative than a scientific discussion. A brief acknowledgment of their contributions, framed within the context of the evolution of hypophosphatasia research, would be more appropriate and align with the expectations of a scholarly publication.

The rest of the manuscript - In its present form, this manuscript is more suitable for publication as a textbook chapter, educational review, or clinical guideline overview rather than a research or review article. The authors should substantially shorten and restructure the paper, provide critical analysis, synthesis, and comparison with existing research, and refocus the introduction and discussion to clearly state aims, highlight novel contributions, and engage with current debates or knowledge gaps.

Figures - Figures 2 and 3 are difficult to interpret and do not add significant value to the manuscript. Figure 2, in particular, is visually hard to read, making it challenging for the reader to extract meaningful information. Additionally, both figures (particularly Figure 3, generated using the STRING database) present data that are publicly and readily accessible online. As such, their inclusion in this form does not contribute original analytical insight or novel visualization.

Comments on the Quality of English Language

The language, though grammatically correct, is redundant in many sections. Streamlining the text for clarity and conciseness would greatly improve readability. 

Author Response

Thank you for the opportunity to revise our manuscript and thank you for your comments and suggestions.

The manuscript presents a comprehensive overview of hypophosphatasia and the ALPL gene, including inheritance patterns, genetic counseling, diagnostic approaches, imaging, and therapeutic strategies. While the paper demonstrates significant effort and includes extensive background information, it does not meet the structural, analytical, and critical standards expected of a publication in a peer-reviewed scientific journal.

Introduction - it does not identify the specific problem or gap that the paper seeks to address. Instead, it gives information about the clinical presentation. The introduction should clearly outline the significance of the topic, current challenges or controversies, and the purpose and scope of the paper.

In the manuscript, the significance of the topic, current challenges or controversies, and the purpose and scope of the paper have been stressed.

Rathbun and Whyte section - I appreciate the author’s effort to acknowledge the seminal contributions of Dr. John Rathbun and Dr. Michael Whyte to the understanding of hypophosphatasia. However, this section is overly detailed and reads more like a biographical narrative than a scientific discussion. A brief acknowledgment of their contributions, framed within the context of the evolution of hypophosphatasia research, would be more appropriate and align with the expectations of a scholarly publication.

This section has been markedly shortened as requested.

The rest of the manuscript - In its present form, this manuscript is more suitable for publication as a textbook chapter, educational review, or clinical guideline overview rather than a research or review article. The authors should substantially shorten and restructure the paper, provide critical analysis, synthesis, and comparison with existing research, and refocus the introduction and discussion to clearly state aims, highlight novel contributions, and engage with current debates or knowledge gaps.

The redundant text has been deleted. This review has been updated with the most informative material available in the literature and is also a comprehensive text that allows the reader to grasp the full knowledge of this topic.

Figures - Figures 2 and 3 are difficult to interpret and do not add significant value to the manuscript. Figure 2, in particular, is visually hard to read, making it challenging for the reader to extract meaningful information. Additionally, both figures (particularly Figure 3, generated using the STRING database) present data that are publicly and readily accessible online. As such, their inclusion in this form does not contribute original analytical insight or novel visualization.

Yes, we agree and both figures have been deleted.

Round 2

Reviewer 3 Report

Comments and Suggestions for Authors

Dear authors,

It appears that the manuscript has not been shortened as previously suggested. In fact, additional information has been added, making the text even longer. For example, the section on the genetic background of the disorder is overly detailed and could be substantially condensed. Same is true for the historical section.

Additionally, the current organization is unclear. For instance, the diagnosis section appears before the epidemiology section, which disrupts the logical flow of the manuscript. The overall structure should be reconsidered and revised to improve readability and coherence.

Author Response

Thank you very much for providing us with the opportunity to revise the manuscript again. We agreed and both genetics and history are shortened, and epidemiology is currently before diagnosis, as suggested.

Many thanks again.
